# Cross-sectional questionnaire study to gather the teaching preferences and expectations of UK undergraduate medical students for culinary medicine learning

Jessica Ying-Yi Xie ,[1] Shoba Poduval ,[1] Victoria Vickerstaff,[1,2] Sophie Park[1]

¹Research Department of Primary Care and Population Health, University College London, London, UK
²Division of Psychiatry, University College London, London, UK

**Correspondence to**
Professor Sophie Park;
Sophie.park@ucl.ac.uk

## ABSTRACT

**Aim** To determine undergraduate medical students' teaching preferences and expectations for Culinary Medicine (CM) learning with a view to informing development of a CM course at a UK medical school.

**Setting** A single, urban UK medical school.

**Participants** 180 undergraduate medical students.

**Study design** A cross-sectional questionnaire study collecting quantitative and qualitative (free-text) data.

**Methods and outcome measures** An online questionnaire consisting of 16 questions of various styles (Likert-type, multiple choice and free-text). Quantitative analysis of multiple choice and Likert-type scale questions was conducted. Qualitative thematic analysis was used to analyse the free-text responses and identify themes.

**Results** Three core themes related to students' understanding of CM were identified: (1) 'CM Learning': students' perceived relevance of CM knowledge, perceived relevance of CM to healthcare and their expectations for teaching; (2) 'The Relationship between Food and Health': links between diet, social factors and health; and (3) 'Evidence-based Medicine': students' perceptions about scientific principles underlying CM. Quantitative analysis revealed that, although 83% of students felt that learning CM is important for their future clinical practice, 56% felt unable to take a dietary history. 73% of students were dissatisfied with the quality, and 78% were dissatisfied with the quantity, of existing medical school teaching understood to be relevant to CM. Topics that students would like to be taught on a CM course included weight management and portion control. Students felt that problem-based style learning would be the most appropriate method for delivering CM teaching.

**Conclusions** This study revealed that medical students felt their dietary counsulting skills could be improved with further clinically relevant teaching in the undergraduate medical curriculum. Students' preferences for CM learning have been taken into consideration in the development of a CM course for fifth-year undergraduate students at a UK medical school, which is delivered during their General Practice placement.

## Strengths and limitations of this study

► The questionnaire was piloted to confirm readability and ensure that questions addressed the study aim.
► At interim analysis of the first 148 questionnaire responses, no new codes were generated during qualitative thematic analysis of free-text responses, signifying that theoretical data saturation had been reached.
► The study was conducted at one medical school only, limiting the transferability of the study findings to other medical schools.
► The response rate (11%) was low which limits the generalisability of the results to the student population.

## INTRODUCTION

Culinary Medicine (CM) may be defined as 'an evidence-based field in Medicine that blends the art of food and cooking with the science of Medicine'.[1] A CM course can offer online learning, didactic teaching and practical hands-on culinary skills in a kitchen setting. The aim is to give students the knowledge and skills to help patients make informed decisions about accessing and eating healthy meals. The General Medical Council (GMC) expects UK medical school graduates to be competent in recognising ill health as a result of poor nutrition, and to be able to apply dietary knowledge to clinical practice.[2] However, it has been reported that medical students and doctors worldwide feel underequipped to provide dietary advice for patients.[3–6] Studies have found that medical students who learn CM are likely to develop positive attitudes towards, and better knowledge of, diet and nutrition; adopt better health habits themselves; and become more competent and confident in dietary counselling.[7–10]

The setting for this study was a single, urban UK medical school, with a six year undergraduate medical (MBBS) curriculum including an Integrated Bachelor of Science (iBSc) in the third year. The first three years are the preclinical part of the MBBS curriculum. In their first year, students receive lectures on nutritional science, such as the digestion of macronutrients and gastrointestinal physiology, and public health nutrition.

The Primary Care Medical Education Team at this UK medical school collaborated with a team of doctors, dietitians and chefs from Culinary Medicine UK[11] to create a CM course that was introduced into the General Practice module of the fifth year of the MBBS curriculum in September 2019. Teaching consisted of online learning and 1 day of face-to-face teaching in a teaching kitchen. The course teaches students culinary skills, evidence-based medicine related to dietary counselling and motivational interviewing skills. Believed to be the first medical school in Europe to provide mandatory CM teaching, the aims of the course are to improve students' confidence in raising the subject of healthy eating and in giving simple dietary advice based on evidence-based recommendations; and to increase students' awareness of potential barriers to healthy eating (for example, cultural norms and low household income). This study was conducted prior to the introduction of the CM course to inform its development by determining students' teaching preferences and expectations for CM learning.

## METHODS

This was a cross-sectional questionnaire study, collecting quantitative and qualitative data using an online (Google Forms) questionnaire. JY-YX (a fourth-year medical student and National Institute for Health Research School for Primary Care Research intern in the University College London (UCL) Research Department of Primary Care and Population Health), led the study design, data collection and analysis.

The study was conducted at a single, urban UK medical school from December 2018 to April 2019.

The study population was 1669 undergraduate medical students enrolled in the academic year 2018–2019 (data supplied by UCL Student Data Analyst).

### The questionnaire

There were a total of 16 questions of various styles (Likert-type, multiple choice and free-text) across four main topics (see table 1). The first topic included establishing students' understanding of the term 'CM'. A free-text response was requested to explore the differences in students' understanding and interpretation of the concept.

At the end of the questionnaire, a free-text box was provided for additional general comments, as well as weblinks for students who wished to learn more about CM.

### Patient and public involvement

Prior to data collection, two medical students (iBSc third-year and MBBS fifth-year) were invited to pilot the questionnaire and comment on the readability. These students suggested rewording of a few questions to improve clarity. The questionnaire was updated accordingly. The lead author is a medical student and contributed to the study design, analysis and writing of this manuscript.

### Data collection

The questionnaire was advertised on posters and via social media and student societies' e-newsletters.

### Analysis

#### Qualitative

NVivo V.12[12] was used to store free-text data in order to conduct a qualitative thematic analysis and identify core themes.[13] JY-YX and ShP independently coded all free-text responses before sharing their interpretations with the qualitative results analysis team (JY-YX, ShP and SP) to discuss any areas of dissonance or contrasting interpretation. Codes were derived deductively from the questionnaire (eg, 'understanding of CM'), and inductively (eg, 'social factors contributing to diet') on data analysis.[14] Careful consideration was given to coding: new codes were compared against preliminary codes and refined, and the data were recoded using the updated set of codes.

| Topic | Information to gather | Question type |
|---|---|---|
| **Table 1** The topic, description and style of question(s) included in the questionnaire | | |
| 1. Education | Students' understanding of the term 'CM' <br> Previous experience(s) of teaching understood to be relevant to CM | Free-text |
| 2. Integrating CM knowledge into clinical practice | Students' self-perceived abilities to facilitate discussions about diet and provide dietary advice for patients | Likert-type |
| 3. Teaching CM | Students' satisfaction with existing medical school teaching understood to be relevant to CM | Multiple choice and Likert-type |
| 4. Features of a CM course | Students' expectations for a CM course for undergraduate medical students, including methods of delivering teaching and teaching topics | Multiple choice |

CM, Culinary Medicine.

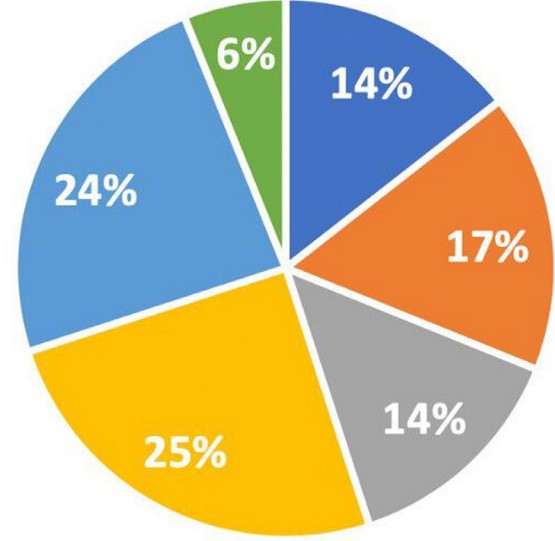

**Figure 1** Results from quantitative analysis of responses to the single choice question: 'Which year of the undergraduate medical programme are you currently in?'. *Students in years 1–3 are grouped under the term 'pre-clinical'. **Students in years 4–6 are grouped under the term 'clinical'.

This improved the quality of analysis.[15] Similar codes were categorised under a subtheme or core theme. No new codes emerged after coding responses from the first 148 students in an interim analysis, suggesting that theoretical data saturation had been reached.

### Quantitative
SPSS Statistics V.25[16] was used to conduct the quantitative analysis. The quantitative results analysis team (JY-YX and VV) carried out descriptive statistics to describe students' year of study. Responses to Likert-type and multiple-choice questions were presented using proportions and frequencies, and responses from preclinical (years 1–3) and clinical (years 4–6) students were compared using $X^2$ and Fisher's exact tests where possible. Statistical significance level was set to 5%.

## RESULTS
### Study population
An 11% response rate (180 responses) was achieved. Fourth-year students were the largest contributors to the study (25% of respondents, see figure 1).

### Students' understanding of CM
There was a wide range of expectations for CM teaching. Three core themes were identified (see table 2).

### Core theme 1: CM Learning
Medical students shared their understanding of CM, suggesting where CM teaching might fit their undergraduate medical training.

#### Subtheme 1: Students' Perceived Relevance of CM Knowledge and Its Application for their Future Clinical Practice
Students shared their expectations to gain clinical knowledge and practical skills from the CM course.

applicable to a clinical session (Third Year).

Learning how to facilitate discussions about diet with patients arose as a key topic.

sensitively debunk health fads (Fifth Year).

conduct a consultation without body shaming (Fifth Year).

Students stated that they would like to be taught how to recognise different eating habits in order to increase their understanding of patients' diet to better facilitate future planning discussions.

understanding different people's feeding behaviours (Second Year).

diet lifestyles, such as veganism (Second Year).

Other students were unfamiliar with the term CM.

**Table 2** Themes and subthemes identified from qualitative thematic analysis of responses to the free-text question; 'What do you understand by the term 'CM'?'

| Core theme | Subthemes |
| --- | --- |
| 1. CM Learning | 1. Students' Perceived Relevance of CM Knowledge and Its Application for their Future Clinical Practice |
| | 2. Students' Perceived Relevance of CM to Healthcare Professions |
| | 3. How to Approach Learning CM |
| 2. The Relationship between Food and Health | 1. Diet and Physical and Mental Health |
| | 2. Social Factors |
| 3. Evidence-based Medicine | |

CM, Culinary Medicine.

first time I came across this term (Fifth Year).

Students who were in favour of a CM course stressed the importance of the topic and considered how the knowledge and skills that they will learn from the course will make them more competent to educate patients about diet and thus improve patients' health outcomes.

this valuable knowledge… is applicable to every specialty (Fifth Year).

CM involves using nutrition to improve patient's health, including teaching patients how to eat and cook healthily (Fourth Year).

However, some students felt that learning CM would not benefit them greatly. Students expressed worry that other topics in the current MBBS curriculum, which they felt were more useful than CM, might be compromised or completely removed.

other parts of the curriculum… should be better developed and prioritised (Third Year).

### Subtheme 2: Students' Perceived Relevance of CM to Healthcare Professional Practice

A variety of perspectives on how CM relates to medical specialities arose.

Students associated CM with nutrition and lifestyle medicine.

related to nutrition (Fifth Year).

part of the wider discipline of 'lifestyle medicine' (Third Year).

Students felt that it may be more appropriate for patients to be counselled about diet by healthcare professionals (HCPs) who have greater clinical expertise in diet and nutrition, rather than doctors.

allied healthcare professionals… are better suited in coaching patients in dietary change, such as dietitians (Fifth Year).

### Subtheme 3: How to Approach Learning CM

There was a wide variety of opinions on the most appropriate methods for teaching undergraduate medical students CM. Students who were enthusiastic about the idea of learning culinary skills, anticipated that practical learning would be engaging and would also allow them to gain knowledge and skills that would enable them to improve their own health and well-being.

a practical kitchen/cooking session… would add variety (First Year).

practical kitchen sessions would provide great insight into how to prepare healthy meals for students as well as future doctors (Fourth Year).

However, other students questioned the feasibility of teaching in a kitchen and whether the skills gained would be useful in future clinical practice.

a practical 'kitchen session' could be very difficult to organise (Fifth Year).

I don't really see how it is practical to teach practical culinary skills [or] what advantage this would confer… we aren't going to be cooking patients' food (Fourth Year).

Some students felt that observing HCPs with expertise in diet and nutrition in clinical practice would be a valuable learning experience.

shadow a dietitian (Fifth Year).

sitting in with diabetic nurses and going through some of the dietary restrictions [for] patients (Fifth Year).

Other students expressed preference for more traditional teaching methods.

a formal tutorial on the… DAFNE [Dose Adjustment For Normal Eating] diet would [be] even more useful (Fifth Year).

Students shared their thoughts on how to integrate CM into the current MBBS curriculum. CM was felt to be closely associated with motivational interviewing, which is covered more broadly elsewhere in the MBBS curriculum outside of the context of diet. Motivational interviewing involves engaging people in exploring their strengths and aspirations regarding a lifestyle habit, evoking motivation to change their behaviour, and supporting them in achieving goals for health and well-being improvement.[17]

how to elicit behaviour change (Second Year).

lifestyle advice [teaching] could be combined with the patient centred pathway teaching in year 5 on motivational interviewing (Fifth Year).

Some students felt that it would be appropriate to dedicate up to 1 day to learning CM and explained that medical students are likely to already have a basic understanding of what constitutes a healthy diet from teaching received elsewhere in the MBBS curriculum.

CM day… any more than that would be excessive. Most students will already know the basics of constructing a good diet (Fifth Year).

There were a range of opinions about which year(s) of the MBBS curriculum the CM course should be delivered in.

integrate preclinical science into FNM ['Fluids, Nutrition and Metabolism' - a module taught in year one of the MBBS curriculum], then teach clinically in year 5 (Fourth Year).

take this module as a final year medical student… as an optional extra module (Fifth Year).

### Core theme 2: The Relationship between Food and Health

Students described their views on the links between diet, social factors and health.

### Subtheme 1: Diet and Physical and Mental Health

Students highlighted the role of diet in preventing, treating and managing disease, and alleviating symptoms due to ill-health.

> nutrition and lifestyle factors associated with diet to both prevent disease and to improve morbidity and mortality (Fifth Year).

Students also recognised mental health problems related to eating.

> eating disorders including anorexia, bulimia, binge eating disorder and other behavioural disorders (Fourth Year).

Students emphasised the need for doctors to recognise the relationships that patients have with food and to promote the health benefits of a balanced diet.

> dieting has been proven to have numerous negative effects on health. [CM] teaching should focus [on] the health benefits of good diet and should not label foods as 'good' or 'bad' as this leads to moral judgement, restriction and reactive bingeing on 'forbidden' foods (Fourth Year).

### Subtheme 2: Social Factors

Students suggested social factors that determine diet and health, including ethnic and cultural background, socio-economic status and level of education.

> level of education… affects your health (Fourth Year).

Students stressed the importance of doctors identifying social factors related to diet and nutrition, such as cultural variation in diets, to enable them to provide patient-centred dietary advice.

> tailor [patients' diets] to lifestyle, ethnic, economic/social factors (Third Year).

> my mum cooks traditional Indian food on a daily basis… [my parents] prefer seeing an Indian GP when they go to the doctors (Third Year).

### Core theme 3: Evidence-based Medicine

Students expressed mixed opinions about the evidence base for CM.

> CM is the evidence-based practice of influencing health through lifestyle changes in food consumption (First Year).

> I'm worried about the quality and the evidence base for CM. Diet and nutrition are… affected by… low-quality research (Fifth Year).

### Integrating CM knowledge into clinical practice

Students' self-perceived confidence and competence related to providing nutritional care were identified (see table 3).

### Provision of CM teaching

Sixty-two per cent of students reported having received no prior teaching perceived to be related to CM (see figure 2).

**Table 3** Results from quantitative analysis of responses to the Likert-type questions. Results are presented as number (percentage)

| Statement | | Preclinical (n=81) | Clinical (n=99) | Total (n=180) | P value |
|---|---|---|---|---|---|
| I am able to facilitate a discussion with patients about diet and/or dietary change. | Disagree | 39 (48.1) | 25 (25.3) | 64 (35.6) | 0.005 |
| | Neutral | 28 (34.6) | 44 (44.4) | 72 (40.0) | |
| | Agree | 14 (17.3) | 30 (30.3) | 44 (24.4) | |
| I am able to take dietary history from a patient. | Disagree | 59 (71.6) | 43 (43.4) | 101 (56.1) | 0.001 |
| | Neutral | 14 (17.3) | 31 (31.3) | 45 (25.0) | |
| | Agree | 9 (11.1) | 25 (25.3) | 34 (18.9) | |
| I am able to apply principles, methods and knowledge relating to nutrition in medical practice and integrate these into patient care. | Disagree | 52 (64.2) | 56 (56.6) | 108 (60.0) | 0.498 |
| | Neutral | 22 (27.2) | 30 (30.3) | 52 (28.9) | |
| | Agree | 7 (8.6) | 13 (13.1) | 20 (11.1) | |
| I am able to provide dietary advice for patients with varying cultural, social and economic needs. | Disagree | 57 (70.4) | 63 (63.6) | 120 (66.7) | 0.617 |
| | Neutral | 18 (22.2) | 26 (26.3) | 44 (24.4) | |
| | Agree | 6 (7.4) | 10 (10.1) | 16 (8.9) | |
| I believe that learning CM is important for my future clinical practice. | Disagree | 6 (7.4) | 4 (4.0) | 10 (5.6) | 0.482 |
| | Neutral | 10 (12.3) | 10 (10.1) | 20 (11.1) | |
| | Agree | 65 (80.2) | 85 (85.9) | 150 (83.3) | |

CM, Culinary Medicine.

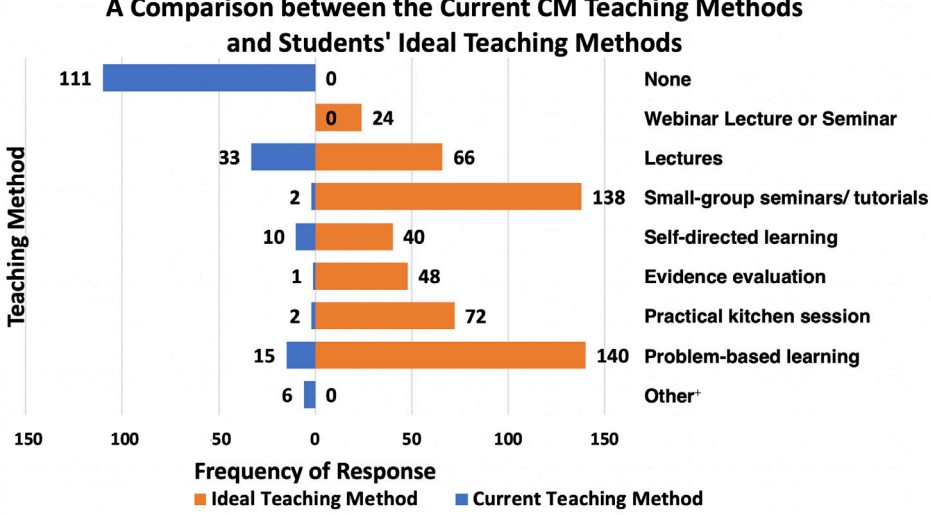

**Figure 2** A comparison of the results from quantitative analysis of responses to two multiple-choice questions: 'What types of CM training have you received so far in the MBBS curriculum?' and 'Select the three most appropriate teaching methods for a CM course'. CM, Culinary Medicine; GP, general practice; MBBS, undergraduate medical degree;

68% preclinical and 77% clinical students were dissatisfied with the quality of existing medical school teaching understood to be relevant to CM. 72% of preclinical and 84% of clinical students were dissatisfied with quantity of existing medical school teaching understood to be relevant to CM (see figure 3).

### Features of CM teaching

Students felt that it is most appropriate for a CM course to include teaching on weight management and portion control (157 responses, see figure 4), closely followed by the types of diet and their evidence base (156 responses). Nutrition psychology was the least popular topic (128 responses).

Students felt that the most appropriate methods of providing CM teaching are problem-based learning (140 responses, see figure 2) and small-group seminars/tutorials (138 responses).

Sixty-seven per cent of students expressed a preference for CM to be taught in preclinical training (preclinical students 72%, clinical students 64%, p=0.257).

### DISCUSSION

Although 83% of students in this study felt that learning CM is important for their future clinical practice, 67% did not feel confident in their knowledge, nor competent to provide specific dietary advice. These findings are similar to that of an Australian study,[5] which found that most preclinical medical students felt that it is important for doctors to have an understanding of nutrition in relation

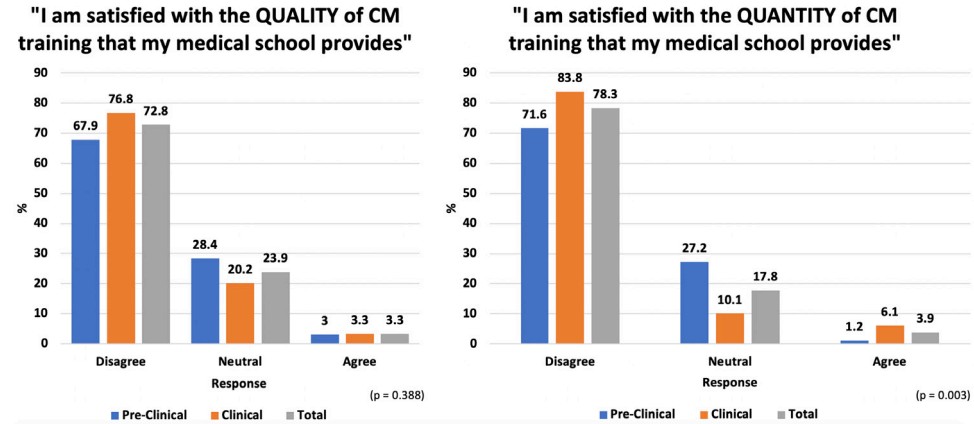

**Figure 3** Results from quantitative analysis of responses to the Likert-type questions: To what extent do you agree with the statement: 'I am satisfied with the QUALITY of CM training that my medical school provides?' and To what extent do you agree with the statement: 'I am satisfied with the QUANTITY of CM training that my medical school provides?'. CM, Culinary Medicine.

## Which Topics and/or Skills do Students Think Should be Included on a CM Course for Undergraduate Medical Students?

**Figure 4** Results from quantitative analysis of responses to the multiple-choice question: 'Which topics and/or skills do you think should be taught on the CM course?'. CM, Culinary Medicine.

to medical conditions, such as coeliac disease, but fewer than half were confident in applying their knowledge to clinical practice.

Some students in this study anticipated that the task of dietary counselling would be delegated to other HCPs, such as dietitians, rather than doctors. The latter views are contradicted by literature that suggests the health benefits for patients when dietary advice is given by doctors, especially in general practice.[18–20] This supports the delivery of CM teaching in the year 5 General Practice placement in the MBBS curriculum. The first aim is to enable students to draw on their experiences of encounters with patients in General Practice and to share these experiences in group discussion about how to approach case-based problems effectively. The second aim is to allow students to apply the dietary counselling skills taught on the CM course to future interactions with patients, including during their ongoing General Practice placement.

The CM course was introduced in September 2019. This medical school now meets the nutrition education recommendation for undergraduate medical students of the European Society for Clinical Nutrition and Metabolism,[21] which recommends the provision of teaching in three domains: basic nutritional science, public health and clinical nutrition.

There may be several reasons why clinical students in this study reported higher levels of self-perceived confidence and competence in their abilities to facilitate discussions about diet and nutrition with patients, compared with preclinical students. First, nutrition teaching outside of CM teaching is delivered in first year of the MBBS curriculum only. During the time that this

questionnaire study was receiving responses (December 2018–April 2019), students in their first year would not yet have received this teaching. Therefore, their knowledge in this area may have been limited. Second, clinical exposure predominantly takes place during years 4–6 of the MBBS curriculum. Senior students therefore have increased patient contact and are more likely to observe HCPs discussing diet and health with patients. Students may have also been offered the opportunity to counsel these patients themselves. Such experiences are likely to have increased clinical students' confidence in this area and may have contributed to the differences in self-reported capabilities between these two groups.

This overall low self-perceived confidence and competence reported by questionnaire respondents may be related to their dissatisfaction with the quality and quantity of pre-existing teaching perceived to be relevant to CM in the MBBS curriculum at the time of this study. This may suggest that this cohort of students had particular interest in, and therefore concern for, undergraduate medical education.

The findings from this study suggest that key components of a CM teaching course placed in the clinical part of an undergraduate medical curriculum are: (1) explaining the clinical and professional relevance of drawing on patient cases from General Practice placements to aid CM learning; and (2) supporting medical students in developing an understanding of how this knowledge might be applied in clinical practice to maximise opportunities for supporting health promotion and managing ill health. These findings were subsequently used to inform the development of a CM course at a UK medical school, as described below.

## Strengths and limitations

A strength of this study is that the questionnaire was piloted by two undergraduate medical students, who checked the readability. The students also confirmed that the questions would enable medical students' teaching preferences and expectations for CM learning to be gathered. This validation step increased the reliability of the questionnaire.

Moreover, JY-YX and ShP conducted the early stages of thematic analysis independently to allow for a wider breadth of interpretations to be captured. The analysis team (JY-YX, ShP and SP) then synthesised and discussed the interpretations before reaching an agreement.

The questionnaire response rate was low (11%) which limits the generalisability of the findings. The low response rate may be due to the fact that CM is a relatively new term in Medicine and is not commonly used among HCPs in the UK and CM is not yet included on most UK medical schools' curricula. The authors anticipated that students with prior understanding and/or interest in CM, were more likely to complete the questionnaire. Therefore, in an attempt to increase the response rate, advertisements for the questionnaire, as well as the participant information sheet, emphasised that all medical students were eligible to take part in the study and that a wide breadth of experiences and perspectives, including no prior understanding of CM, were welcome. Even with this additional encouragement, the response rate remained low. However, theoretical saturation of the qualitative data was judged to have been reached after coding the first 148 responses in an interim analysis, implying that sufficient qualitative data had been gathered for the views of students to be represented in the free-text responses. This strengthened the credibility of the study.

Another limitation is that questionnaire responses were received from one UK medical school only. Factors such as differences between medical school curricula may limit the transferability of the study findings to other medical schools.

## Implications

The findings from this study have been used to inform the development of a CM course for undergraduate students at a UK medical school during their year five General Practice placement.

## CM course content

The course includes a wide range of teaching topics. The aim is for teaching to meet the learning needs of students, which were identified in this study, as well as GMC[2] requirements for nutrition education. The course topics are weight management and portion control, nutritional screening tools, taking a dietary history, motivational interviewing skills, addressing the dietary needs of patients with varying cultural and ethnic backgrounds, types of diet and their evidence-base, and practical culinary skills and food preparation.

## CM teaching methods

The teaching methods that students in this study expressed preferences for have been included in the course to optimise their learning potential. These teaching methods include an online module, a face-to-face tutorial, case-based discussions, motivational interviewing role-play and culinary skills training. The latter has been suggested to be an effective teaching method to improve students' nutrition knowledge and food identification skills.[8]

The course creators prepared patient cases for discussion to maximise clinical relevance. Students also prepare their own summaries of relevant patient cases encountered during their General Practice placement, and are supported by course facilitators to think about how they will apply the principles from CM teaching in future consultations.

## Integrating CM teaching into the clinical curriculum

CM teaching has been integrated into the clinical curriculum to enable students to have a basic grounding in the knowledge and practical skills of CM, which are highly relevant to all healthcare professions and patient care.

## Future research in this area

The questionnaire that was created for this study has been adapted into a post-course questionnaire and is being used to gather students' feedback on their CM learning experiences. The feedback data will be analysed in a separate study.

Conducting this cross-sectional questionnaire study at other medical schools will enable faculties to determine whether there is a need to introduce or improve CM teaching at their medical school to better meet their students' needs and expectations for CM learning and/or GMC[2] requirements for nutrition education.

Future qualitative research in this area will allow deeper exploration of students' thoughts regarding CM teaching.

## CONCLUSION

This cross-sectional quantitative study has provided insight into medical students' understanding of and attitudes towards learning CM. The study has highlighted students' perceptions of how they feel CM knowledge and skills might help them to meet the GMC's Outcomes for Graduates and fulfil their future role as doctors. Students reported a lack of confidence in their dietary counselling skills and felt that they could benefit from further integrated and clinically relevant teaching. Students' preferences for CM teaching have been taken into consideration in the development of a CM course for fifth-year undergraduate students at a UK medical school during their General Practice placement. This medical school is an exemplar of how CM teaching can be introduced into the clinical component of an undergraduate medical curriculum.

**Acknowledgements** The authors acknowledge Professor Ann Griffin, Professor Elizabeth Murray, Miss Valeriya Kopanitsa, Dr Deepash Hosadurg, the National

Institute for Health Research School for Primary Care Research (NIHR SPCR) and UCL Student Data Analyst. They are thankful to all the study participants and to Dr Rupy Aujla, Elaine Macaninch and colleagues at Culinary Medicine UK for their support in setting up the course.

**Contributors**  JY-YX—main author and lead for the study design, participant recruitment, data collection, and analysis and write-up. SP—chief investigator, co-supervisor and contributed to the study design, data analysis and write-up. ShP—co-supervisor and contributed to the study design, participant recruitment, data analysis and write-up. VV—supervised the statistical analysis and write-up.

**Funding**  This work was supported by the National Institute for Health Research School for Primary Care Research (NIHR SPCR) grant number 156780. VV receives Seedcorn Funding from the NIHR SPCR.

**Competing interests**  None declared.

**Patient consent for publication**  Not required.

**Ethics approval**  Ethical approval to carry out this study was received from UCL Research Ethics Committee (ref.12471/002). Study participants were asked provide informed consent by reading the participant information sheet (PIS) on the first page of the questionnaire and confirming that they fully understood their role in the study and how the data collected would be used. Once consent had been gained, participants were able to proceed to the questionnaire on the next page. Questionnaires were completed anonymously. No participant-identifiable data were collected. The only personal data collected were year of study, but participants could not be identified from the information that they provided.

**Provenance and peer review**  Not commissioned; externally peer reviewed.

**Data availability statement**  All data relevant to the study are included in the article or uploaded as supplemental information. The data generated from this study are not suitable for sharing beyond that contained within the report. Further information can be obtained from the corresponding author.

**ORCID iDs**
Jessica Ying-Yi Xie http://orcid.org/0000-0003-1490-7765
Shoba Poduval http://orcid.org/0000-0002-3049-123X

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
