## [Reviewer comments · BMJ Open]

ARTICLE DETAILS

TITLE (PROVISIONAL)	A Cross-sectional Questionnaire Study to Gather the Teaching Preferences and Expectations of UK Undergraduate Medical Students For Culinary Medicine Learning
AUTHORS	Xie, Jessica; Poduval, Shoba; Vickerstaff, Victoria; Park, Sophie

VERSION 1 – REVIEW

REVIEWER	Michelle Hauser Stanford University, USA
REVIEW RETURNED	22-Jan-2020

GENERAL COMMENTS	Comments on BMJ review of “What Are the Preferences and Expectations of UK Undergraduate Medical Students From a Culinary Medicine Course?” Overall, the evaluation and creation of effective CM courses is an important topic and I am glad to see the authors focusing on this. A lot of interesting information was gathered, and appropriate qualitative methods were used. The data presented are of interest to those in building academic programming for HCPs in training and the topic is timely. However, I recommend major revision prior to being accepted for publication. This is based on the following concerns that I had in reviewing the paper: • It seems as though most students responding to the questionnaire had no CM lessons to date as from the authors’ limited description, it sounds as if what is included occurs late in training. The responses to all questions and their display in tables and figures, as well as statistical analysis, needed to be separated by those who had and had not had exposure to CM course content. Without this separation throughout, results are difficult to interpret.• The manuscript would have benefited from a more detailed description of the CM course offerings currently available to students at the school; this should go in the methods section.• It would have been easier to review the manuscript and analyses had a copy of the questionnaire been provided as a supplement• It seemed the purpose of this study was to inform the creation of new CM content to be offered. While some data was gathered on topics students felt was important, it seems this question quantitative and without a free text/qualitative opportunity for students to express other topics of interest. This is a limitation of the study and in the application of the results to their intended goal.• Lines 10-13 p.5 state “We aimed to obtain 313 survey responses for our results to be statistically significant at the 95%
---

	confidence interval with a 5% margin for error.” It is unclear what exactly was the aim of achieving statistical significance and/or what would be measured to determine significance. This was a questionnaire evaluation rather than an interventional study. There was no need to achieve significance. It would have been better to design the study and student outreach to get the highest possible response rate (which was low at 11%) in order to make sure that results were applicable to the student population. Furthermore, lines 50-53 on p.13 describe non-significant results as a limitation. This is not a limitation and not an effective interpretation of statistical significance in the context of this study/evaluation. It would be better to include in the discussion more commentary on why some responses were significantly different b/w clin and preclin groups and why others were not. I actually found it interesting when there was no significant difference as this meant that preclinical and clinical students agreed, whereas those items that were statistically significant were points of disagreement between groups and worth delving into.  • Need to describe how it was determined that theoretical saturation was achieved • It is unclear to me why basic demographic characteristics were not collected as part of this evaluation as this could be done without identifying participants, especially for gender. It would lend to generalizability to know the race/ethnicity breakdown of respondents. • For the question that is the basis for Table 2, “What do you understand by the term ‘CM’?”, this seems too vague to answer. Maybe this was the authors’ intention? If so, please describe in-text. If not their intention, I think this issue would have been identified if a larger pilot group were used at the outset. • Beginning on p.7 under the heading “CM Learning” some of the phrases included are too brief to lend support to the authors’ interpretations. It would be helpful to have a bit more context for some of them. This seemed to improve as the section on qualitative analysis results progressed. • While the authors clearly point out translatability as a limitation due to including only one medical school, there are other limitations to translatability. 1) current curriculum not described (as mentioned above), so it is unclear how results would apply to other locales. 2) The student population that responded to the questionnaire is more aware of CM than the general population of any group of medical students I have encountered. Given the low response rate, those that responded may be more likely than average to know and have opinions about CM in the curriculum than those who did not. • Under “Provision of CM Training” in the Results, this isn’t an effective evaluation of current content/sessions since it’s unclear how many/what proportion of those saying the quality is poor have actually attended any CM session. Based on the numbers/other results, it seems many probably did not. Those who haven’t had CM sessions shouldn’t be evaluating the quality of the sessions. These responses should be non-applicable, keeping only responses from those who reported CM sessions/content exposure in the curriculum. • For Figure 1, rather than including just clinical and preclinical designations, it would be helpful to show those who had/n’t had CM content exposure in the curriculum • There are some grammatical and punctuation errors throughout
--	---

	If the authors are able to address the majority of the above concerns, I do think this warrants publication as it is a unique contribution to the literature on a timely topic. Thank you for the interesting read and for the invitation to review.
--	--

REVIEWER	Ala Shaikhkhalil, MD Nationwide Children's Hospital/The Ohio State University, USA I reviewed this manuscript along with a trainee, Dr. Shukla-Udawatta. We both discussed the article and delivered review while maintaining confidentiality.
REVIEW RETURNED	12-Apr-2020

GENERAL COMMENTS	Overall, this is an important topic and, this clearly required an effort. So congratulations! My main critiques  • You mention having designed and conducted this survey to inform the implementation of CM in your school, but I cannot find enough (or any) information about how the results of this survey did indeed inform your CM implementation. • Further, can you give an outline of your current CM course? How many modules? How is it delivered? • In many instances, you mention that “many”, “few”, “minority” of the students. Would like to see actual percentages or numbers • Below is specific section by section feedback Introduction:  1. You are talking about a course that was developed with input from CM experts. Can you give more info on those experts (their discipline, same institution?) and the actual course? Methods:  2. The calculation of sample size- was that done to look at quantitative or qualitative data or both? Can you show us how you came up with numbers? 3. Data coding: Could you give examples of the predefined codes for deductive coding and what codes came out from the inductive coding. (You have some in results, but consider giving in methods as well). Results  4. In multiple places in the manuscript” Define what “many students” means, with numbers or percentages of study participants. Not enough to say”many students”. 5. Were the students who had comments suggesting that CM curriculum would not be helpful more or less likely to report comfort with nutrition counseling? 6. This may be too simplistic: but can you show how you know that you reached theoretical saturation? 7. “56% felt unable to take a dietary history” this conflates dietary history with CM. Someone can take a great dietary history without CM. Did you specify the ability to take a history and give counseling in food? 8. “73% of students were dissatisfied with the quality, and 78% were dissatisfied with the quantity, of CM training”- since the school did not have CM education, how did you measure dissatisfaction? Did you mean nutrition education? 9. In abstract, nothing about method of delivery or setting, even though the conclusions cite that Discussion:  10. Page 12: Define “some students positioned CM as highly relevant to patient care.” How many students? Also include similar
---

	numbers in the next paragraph beginning with “some students perceived CM as relevant to clinical practice with patients.” 11. Did you collect data on satisfaction after the course? How long has the course been in place? 12. Give more details about your current course in this section or in the implications section? 13. Importantly, how DID the survey results inform the design and implementation of the curriculum. Implications: 14. Are you able to provide data about how your CM course delivered education (small groups versus hands on sessions) 15. Is there any data about who does/should deliver these courses? Physicians or dieticians, or collaboration of both? Conclusions: 16. How feasible is it to arrange for small group teaching about diets and have hands on culinary skills teaching as opposed to the current methods online? Article Summary 17. Please separate strengths and limitations. 18. “The questionnaire was piloted to confirm that the questions fulfilled the study aim. “ clarify the study aim and what is meant by piloted 19. “All students enrolled at a UK medical school (academic year 2018-19) were invited to take part in this study to gather a wide breadth of views and experiences of CM “. I do not think that this is a strength. 20. ” JX and ShP independently analyzed the free-text responses to gather different perspectives on the data. “is this a strength or limitation? Why?
--	--

REVIEWER	Sandra Nicholson QMUL UK
REVIEW RETURNED	10-May-2020

GENERAL COMMENTS	Thank you for submitting an evaluation of a course that is data packed. Please find my suggestions for improvement which I feel are essential before publication is possible. The abstract needs an introductory sentence defining culinary medicine. The abstract further lacks any background as to why this is of importance to an undergraduate medical degree. The introduction would be improved by additionally explaining the difference/synergy between cooking, diet, nutrition and health/ill health and medicine. In addition whilst there was a sub-title "article summary" this appears to have been omitted from the draft unless the subsequent paragraph outlining the strengths and weaknesses accounts for this?? Why do you need to open with a summary? How does this differ from the abstract? Methods Is there something different about the students who signed up? How representative are they? You comment that more of them are from the 4th year.
--

More detail about how the questionnaire was designed and perhaps including the questionnaire as an appendix would be helpful.

Piloting the questionnaire with 2 medical students prior to dissemination is a good idea but I struggle to see how this is PPI?

Results

For me this was the most problematic section and needs the most clarification and revision. It would be helpful to have an introduction stating how the results are to be presented to orientate the reader. For example I am clear what the 3 core themes pertaining to students' understanding of CM are but had to work out for myself where this data came from (topic 1). As a minimum the results section needs introductory sentences to each topic results as it cannot be expected that the reader will skip back and forth in the text to find this out.

p.7 lines 3-4 "majority of students" this is the majority of those that replied to the questionnaire and should be written as such please. I suggest you need to decide on a standardised system for reporting your results to avoid duplication of findings. For example there is no need to have table 3 and repeat percentages again in text. I suggest you have the table in the results and either include a very short interpretation of this or discuss the implications in discussion section.

However you have chosen not to include certain figs such as fig 2 in the main body of your paper and instead have provided a summary of its message. Please clarify why? And ensure consistency of presentation.

Overall I do wonder if you are trying to present too much data, and maybe the results section could be simplified, highlighting your main finding, which might actually strengthen your main messages, some of which I feel are lost.

Discussion

This is an important section for your paper as it presents the opportunity to integrate your findings into a coherent message. I think the first paragraph needs better structuring. This could be achieved by a short overall summary of what you consider is the most important findings, then go on to describe how these integrate with the literature. I would consider how to re-phrase sentences with "many" and "some" as descriptors of student responses as these are generally unhelpful.

Much of the conclusion section reads as a summary and could be better placed at the start of the discussion section as suggested. This would then leave the writers opportunity to highlight what can be concluded from their overall findings. I suggest to the writers that they consider amalgamating the most important issues from the implications and current conclusions to present a sharper more concise take home message for readers.

VERSION 1 – AUTHOR RESPONSE

Reviewer 1

Reviewer 2

	Comment	Authors' Response and Rationale
1	You mention having designed and conducted this survey to inform the implementation of CM in your school, but I cannot find enough (or any) information about how the results of this survey did indeed inform your CM implementation.	In the 'Implications' section, we have made a series of recommendations for the development of the CM course at UCLMS and what future research in this area might look like, based on the findings from this study.
2	Further, can you give an outline of your current CM course? How many modules? How is it delivered?	Please see  • Reviewer 1, Comment 2 • Reviewer 2, Comment 4
3	In many instances, you mention that "many", "few", "minority" of the students. Would like to see actual percentages or numbers	We have removed these descriptions of qualitative data as we feel it is not possible to quantify how many students expressed a certain opinion/ suggestion/ experience, and because we feel that the free-text data from each and every one of the participants is valuable and important. We do not want to suggest that a certain opinion/ suggestion/ experience is of less importance to significance compared alternative opinions/ suggestions/ experiences, as this is qualitative data.
Introduction		

4	You are talking about a course that was	In the introduction, we have added a more detailed description of the CM experts and CM course at UCL.
---	---	--

	Comment	Authors' Response and Rationale
1	It seems as though most students responding to the questionnaire had no CM lessons to date as from the authors' limited description, it sounds as if what is included occurs late in training. The responses to all questions and their display in tables and figures, as well as statistical analysis, needed to be separated by those who had and had not had exposure to CM course content. Without this separation throughout, results are difficult to interpret.	Students had not received any formal Culinary Medicine (CM) training as part of their undergraduate medical training when they took part in our study. The only nutrition teaching that students received prior to the introduction of the CM course was lectures on nutrition from the public health perspective and nutritional science. A statement clarifying this has been added to the introduction. A description of what a CM course entails can be found in the introduction. A CM course was introduced at University College London (UCL) Medical School (MS) in September 2019 and our study was completed prior to the commencement of the course. Students responded to the questionnaire based on teaching experiences (or lack of) that students understood to be relevant to CM; the authors did not provide students with a definition of, or any background information about, CM and CM training. However, a few links to online resources were provided at the end of the questionnaire for students who might want to read around the subject after completing the questionnaire. Figure 2 suggests that students' experience of teaching understood to be relevant to CM was minimal - the highest number of responses (111) were recorded for having received no CM teaching from UCLMS and, where students reported to have received teaching that they understood to be relevant to CM, their experience was extremely varied (e.g. only 1 student claimed to have been taught evidence evaluation). Therefore, we felt that there would be little to infer from presenting the results as a comparison between students who reported receiving teaching relevant to CM from UCLMS with students who reported that they have not received teaching relevant to CM from UCLMS. When analysing the qualitative data, we chose to compare pre-clinical and clinical students' experiences and views on teaching relevant to CM to see if greater clinical exposure and interaction with patients in clinical years (years 4 – 6) changes students' opinion on the importance and clinical significance of CM to their future practice. Students on

		their clinical placements observe a higher number of consultations during which clinicians discuss diet and nutrition with their patients, compared to in pre-clinical medical training (years 1 – 3). Where our p-values were <0.05, it showed that greater clinical exposure did not change students' opinions, and vice versa for where p-values were >0.05.
2	The manuscript would have benefited from a more detailed description of the CM course offerings currently available to students at the school; this should go in the methods section.	As mentioned in Reviewer 1, Comment 1, UCLMS had not provided any CM training for students prior to when this study was conducted. The aim of the study was to determine student preferences and expectations of learning about CM. A statement clarifying the nutrition teaching that students receive in pre-clinical training has been added to the introduction. In the introduction, readers are familiarised with the concept of CM using a description from the current literature of what CM teaching involves. The CM course introduced at UCLMS after the completion of the survey is also described in the introduction (rather than the methods section), in order to clarify that the survey was not an evaluation of an existing CM course, but an evaluation of student preferences and expectations used to inform the development of the course.
3	It would have been easier to review the manuscript and analyses had a copy of the questionnaire been provided as a supplement	A copy of the questionnaire will be provided as a supplementary document when the revised manuscript is submitted.
4	It seemed the purpose of this study was to inform the creation of new CM content to be offered. While some data was gathered on topics students felt was important, it seems this question quantitative and without a free text/qualitative opportunity for students to express other topics of interest. This is a limitation of the study and in the application of the results to their intended goal.	An 'Other' option, with a free-text box was provided for this question (see question 17 of the copy of the questionnaire). However, none of the questionnaire respondents selected 'other' and therefore did not write any comments in the free-text box. As there were no responses for this option, a p-value could not be calculated. We acknowledge that figure 4 contained an error so that it did not display that students did not select the option 'other' to question 17. We have corrected this error.
5	Lines 10-13 p.5 state "We aimed to obtain 313 survey responses for our results to be statistically significant at the 95% confidence interval with a 5% margin for error." It is unclear what exactly was the aim of achieving statistical significance and/or what would be measured to determine	Thank you for the comments about response rate. We agree that a high response rate was important. As described in the 'Methods' section, we used multiple methods of advertisement in an effort to increase the questionnaire response rate, whilst adhering to UCL Medical School policy regarding advertising our survey (namely we could not contact students by emailing the university email addresses of all 1669 undergraduate students in the academic year 2018-19). The response

	significance. This was a questionnaire evaluation rather than an interventional study. There was no need to achieve significance. It would have been better to design the study and student outreach to get the highest possible response rate (which was low at 11%) in order to make sure that results were applicable to the student population. Furthermore, lines 50-53 on p.13 describe non-significant results as a limitation. This is not a limitation and not an effective interpretation of statistical significance in the context of this study/evaluation. It would be better to include in the discussion more commentary on why some responses were significantly different b/w clin and preclin groups and why others were not. I actually found it interesting when there was no significant difference as this meant that preclinical and clinical students agreed, whereas those items that were statistically significant were points of disagreement between groups and worth delving into.	rate was not limited by the original sample size calculation. We have decided to remove the sample size calculation given it was a questionnaire and there was no need to reach statistical significance. Equally, we achieved saturation for the qualitative work with the sample obtained. We have now removed the section describing the non-significant results as a limitation. Commentary on why responses might be different between clinical and pre-clinical students has been added to the 'Discussion' section.
6	Need to describe how it was determined that theoretical saturation was achieved	An explanation of how theoretical saturation was achieved has been added in the 'Analysis' section, under the sub-heading 'Qualitative'.
7	It is unclear to me why basic demographic characteristics were not collected as part of this evaluation as this could be done without identifying participants, especially for gender. It would lend to generalizability to know the race/ethnicity breakdown of respondents.	We did not collect basic demographic characteristics because it was not relevant to our study aims and objectives. We aimed to achieve generalizable results with a reasonable response rate to the survey. Further comparisons, such as those suggested by, could be investigated in future research.
8	For the question that is the basis for Table 2, "What do you understand by the term 'CM'?", this seems too vague to answer. Maybe this was the authors' intention? If so, please describe in-text. If not their intention, I think this issue	We intended for the first question in the questionnaire to be an open question to allow students to express their understanding, if any, of the term 'CM'. We found that some students had never heard of this term before, whilst others had some understanding. We have added an explanation for this to the methods section. A copy of

	would have been identified if a larger pilot group were used at the outset.	the questionnaire will be provided as a supplementary document when the revised manuscript is submitted
9	Beginning on p.7 under the heading “CM Learning” some of the phrases included are too brief to lend support to the authors’ interpretations. It would be helpful to have a bit more context for some of them. This seemed to improve as the section on qualitative analysis results progressed.	Some of the student responses were short due to this being a questionnaire rather than an interview study. We agree that more context would be helpful for readers and have tried to add context by providing short statements introducing each response.
10	While the authors clearly point out translatability as a limitation due to including only one medical school, there are other limitations to translatability. 1) current curriculum not described (as mentioned above), so it is unclear how results would apply to other locales. 2) The student population that responded to the questionnaire is more aware of CM than the general population of any group of medical students I have encountered. Given the low response rate, those that responded may be more likely than average to know and have opinions about CM in the curriculum than those who did not.	Please see Reviewer 1, Comment 1 We have a statement in the ‘Strengths and Limitations’ section stating that we recognised that the questionnaire respondents may have been more familiar with the term CM than those who did not answer the questionnaire. However, we have rephrased this for clarity/ further emphasis.
11	Under “Provision of CM Training” in the Results, this isn't an effective evaluation of current content/sessions since it's unclear how many/what proportion of those saying the quality is poor have actually attended any CM session. Based on the numbers/other results, it seems many probably did not. Those who haven't had CM sessions shouldn't be evaluating the quality of the sessions. These responses should be non-applicable, keeping only responses from those who reported CM sessions/content exposure in the curriculum.	Please see Reviewer 1, comment 1. UCLMS did not provide any formal CM training before or when this study was conducted. Therefore, students were asked about their experience of any teaching which they understood to be relevant to CM at UCLMS (such as formal nutrition teaching). Most students identified that there was a lack of/ insufficient teaching relevant to CM, as stated under ‘Provision of CM Training’.
12	For Figure 1, rather than including just clinical and preclinical designations, it would be helpful to show those who had/n't had CM content exposure in the curriculum	Please see Reviewer 1, Comment 1

13	There are some grammatical and punctuation errors throughout	The manuscript has been thoroughly checked and re-checked for errors and these have been corrected
	developed with input from CM experts. Can you give more info on those experts (their discipline, same institution?) and the actual course?	
Methods		
5	The calculation of sample size- was that done to look at quantitative or qualitative data or both? Can you show us how you came up with numbers?	Please see Reviewer 1, Comment 5
6	Data coding: Could you give examples of the predefined codes for deductive coding and what codes came out from the inductive coding. (You have some in results, but consider giving in methods as well).	Example of predefined codes have been added under 'Analysis' in the 'Methods' section.
Results		
7	In multiple places in the manuscript" Define what "many students" means, with	Please see Reviewer 2, Comment 3

	numbers or percentages of study participants. Not enough to say "many students".	
8	Were the students who had comments suggesting that CM curriculum would not be helpful more or less likely to report comfort with nutrition counseling?	Summary of results: The only (1 out of 180) student that felt that CM is very unimportant (1/5) to their future clinical practice, expressed complete satisfaction (5/5) with the quality and quantity of current teaching understood to be relevant to CM, yet felt completely incapable of facilitating a discussion with patients about diet and/or dietary change (1/5), applying principles and integrating into patient care (1/5), providing dietary advice (1/5), taking a dietary Hx (1/5). They did not provide any comments in the free-text boxes to explain this. 9 out of 180 students felt that CM is unimportant (2/5) to their future clinical practice and had a median score 3/5 for self-perceived ability to facilitate dietary counselling ecetera, and for the quality and quantity of current CM training, implying that they neither agree nor disagree. It is therefore unlikely that students who did not think CM learning would be helpful did so because they already felt capable of providing nutrition counselling. However, the numbers are too small for us to include this in the main findings of the paper.
9	This may be too simplistic: but can you show how you know that you reached theoretical saturation?	Please see Reviewer 1, Comment 6
10	"56% felt unable to take a dietary history" this conflates dietary history with CM. Someone can take a great dietary history without CM. Did you specify the	As displayed in Table 3, we asked students to what extent they agree with the following statements: 'I am able to facilitate a discussion with patients about diet and/or dietary change', 'I am able to take dietary history from a patient', 'I am able to apply principles, methods and knowledge relating to nutrition to medical practice and integrate these into patient care' and 'I am able to provide dietary advice for patients with varying cultural, social and economic needs'. Therefore, we specified the ability to take a history and give dietary counselling

	ability to take a history and give counseling in food?	
1 1	“73% of students were dissatisfied with the quality, and 78% were dissatisfied with the quantity, of CM training”- since the school did not have CM education, how did you measure dissatisfaction? Did you mean nutrition education?	We allowed students to answer the survey based on their understanding of CM. Some students interpreted this as nutrition and bioscience, whilst others felt it encompasses consultation skills, like motivational interviewing. Figure 2 displays how many responses were recorded for CM training delivered by UCLMS. Students answered this question based on their level of satisfaction with the quality and quantity of CM training delivered by UCLMS, whereby ‘CM training’ is open to the interpretation of students
1 2	In abstract, nothing about method of delivery or setting, even though the conclusions cite that	Please see  • Reviewer 1, Comment 2 • Reviewer 2, Comment 1
Discussion		
1 3	Page 12: Define “some students positioned CM as highly relevant to patient care.” How many students? Also include similar numbers in the next paragraph beginning with “some students perceived CM as relevant to	Please see Reviewer 2, Comment 3

	clinical practice with patients.”	
14	Did you collect data on satisfaction after the course? How long has the course been in place?	Our study was completed before CM training was introduced at UCL in September 2019. A separate team in the UCL Research Department of Primary Care and Population Health team have conducted a separate study to gather post-course feedback from students. The course has been in place since September 2019. This has been clarified in the introduction.
15	Give more details about your current course in this section or in the implications section?	Please see Reviewer 2, Comment 4
16	Importantly, how DID the survey results inform the design and implementation of the curriculum.	Please see Reviewer 2, Comment 1
Implications		
17	Are you able to provide data about how your CM course delivered education (small groups versus hands on sessions)	Descriptions of the CM course and how it could be developed (i.e study implications) have been provided. Please see  • Reviewer 1, Comment 2 • Reviewer 2, Comment 1
18	Is there any data about who does/should deliver these courses? Physicians or dieticians, or collaboration of both?	The CM course at UCLMS was introduced in September 2019 by the UCL Primary Care team in collaboration with Culinary Medicine UK. Course facilitators consist of dietitians (to provide the evidence-based medicine), GPs (to provide expertise in providing patient-orientated care) and a chef (teaches the practical culinary skills). Details have been included in the introduction.
19	How feasible is it to arrange for small group	The UCLMS CM course is currently being evaluated and the feasibility of the delivery methods will be discussed as part of future research.

	teaching about diets and have hands on culinary skills teaching as opposed to the current methods online?	
Article Summary		
20	Please separate strengths and limitations.	The BMJ Open requires “ An Article Summary, placed after the abstract, consisting of the heading ‘Strengths and limitations of this study’ ”(please see here for reference: https://bmjopen.bmj.com/pages/authors/#reporting_patient_and_public_involvement_in_research). Therefore, we have not separated strengths and limitations in the article summary. However, ‘Strengths and limitations’ further down in the manuscript, have been separated, presenting strengths first, and then limitations.
21	The questionnaire was piloted to confirm that the questions fulfilled the study aim. “clarify the study aim and what is meant by piloted	We have clarified our descriptions of the ‘pilot’ as follows:  • Under ‘Strengths of This Study’ under the ‘Article Summary’ section • Under ‘Stakeholders’ under the ‘Methods’ section • Under ‘Strengths and Limitations’
22	“All students enrolled at a UK medical school (academic year 2018-19) were invited to take part in this study to gather a wide breadth of views and experiences of CM “. I do not think that this is a strength.	We invited all students in order to avoid purposively sampling students (e.g. according to their experiences, interest in nutrition/ Lifestyle Medicine, or by their year group) and representing the views and experiences of one group of students. We therefore see this as a strength, as our results are more representative of the opinions and experiences of students at UCLMS in 2018/19. We have updated the strengths and limitations section accordingly.
23	” JX and ShP independently analyzed the free-text responses to gather different	This is a strength and we have reworded this point.

	perspectives on the data. "is this a strength or limitation? Why?"	
--	---	--

Reviewer 3

	Comment	Authors' Response and Rationale
1	The abstract needs an introductory sentence defining culinary medicine. The abstract further lacks any background as to why this is of importance to an undergraduate medical degree.	We have included a definition of CM and explanation of the importance of the subject in the 'Introduction' section and feel that this is not required in the abstract
2	The introduction would be improved by additionally explaining the difference/synergy between cooking, diet, nutrition and health/ill health and medicine.	We have explained the synergy between diet and health in throughout the 'Introduction' section
3	In addition whilst there was a sub-title "article summary" this appears to have been omitted from the draft unless the subsequent	For original research submissions, the BMJ Open requires "An Article Summary, placed after the abstract, consisting of the heading 'Strengths and limitations of this study'", and containing up to five short bullet points, no longer than one sentence each, that relate specifically to the methods. They should not include the results of the study." (https://bmjopen.bmj.com/pages/authors/). We have followed the journal's requirements.

	paragraph outlining the strengths and weaknesses accounts for this?? Why do you need to open with a summary? How does this differ from the abstract?	
Methods		
4	Is there something different about the students who signed up? How representative are they? You comment that more of them are from the 4th year. More detail about how the questionnaire was designed and perhaps including the questionnaire as an appendix would be helpful.	Under ‘Study population’ of the ‘Results’ section, we have stated ‘see Figure 1’, which was uploaded as a separate file, as required by BMJ Open. Figure 1 shows a pie chart of the breakdown of the year groups of students (years 1 – 6) as percentages of 180 (the total number of questionnaire respondents and therefore total number of students who participated in our study). Please see Reviewer 1, comment 3 A copy of the questionnaire will be provided as a supplementary document when the revised manuscript is submitted.
5	Piloting the questionnaire with 2 medical students prior to dissemination is a good idea but I struggle to see how this is PPI?	The BMJ Open requires a subtitle ‘Patient and Public Involvement’ in the methods section (please see here for reference: https://authors.bmj.com/policies/patient-public-partnership/; and https://bmjopen.bmj.com/pages/authors/#reporting_patient_and_public_involvement_in_research . We used these examples, as recommended by the BMJ, as a guide to write the PPI statement: https://docs.google.com/document/d/1VQoMQfPuCPds8S9MehBUF50oi-ZQOrH0gmwd6D_PuxU/edit?ts=5a7c7621. We have followed the journal’s requirements.
Results		

6	For me this was the most problematic section and needs the most clarification and revision. It would be helpful to have an introduction stating how the results are to be presented to orientate the reader. For example I am clear what the 3 core themes pertaining to students' understanding of CM are but had to work out for myself where this data came from (topic 1). As a minimum the results section needs introductory sentences to each topic results as it cannot be expected that the reader will skip back and forth in the text to find this out.	We have added "Core theme 1/2/3" at the start of each core theme in the body of the results section. We have added "Sub-theme 1/2/3" at the start of each sub-theme in the body of the results section. There is an introductory sentence under each presented core theme.
7	p.7 lines 3-4 "majority of students" this is the majority of those that replied to the questionnaire and should be written as such please.	Please see Reviewer 2, Comment 3

8	I suggest you need to decide on a standardised system for reporting your results to avoid duplication of findings. For example there is no need to have table 3 and repeat percentages again in text. I suggest you have the table in the results and either include a very short interpretation of this or discuss the implications in discussion section.	The text describing table 3 has been removed and aspects have been integrated into the 'Discussion' section.
9	However you have chosen not to include certain figs such as fig 2 in the main body of your paper and instead have provided a summary of its message. Please clarify why? And ensure consistency of presentation.	For original research submissions, the BMJ Open requires "that you upload your figures as separate files rather than embedding them in the manuscript." (https://bmjopen.bmj.com/pages/authors/). No figures were embedded in the text. We have followed the journal's requirements.
10	Overall I do wonder if you are trying to present too much data, and maybe the results section could be simplified,	The 'Results' section has been simplified by removing duplications (please see Reviewer 3, Comment 8)

	highlighting your main finding, which might actually strengthen your main messages, some of which I feel are lost.	
Discussion		
1 1	I think the first paragraph needs better structuring. This could be achieved by a short overall summary of what you consider is the most important findings, then go on to describe how these integrate with the literature. I would consider how to re-phrase sentences with “many” and “some” as descriptors of student responses as these are generally unhelpful.	The first paragraph of the ‘Discussion’ section has been re-written. Please see Reviewer 3, Comment 11. Please see Reviewer 2, Comment 3
1 2	Much of the conclusion section reads as a summary and could be better placed at the start of the discussion section as suggested. This would	Elements of what was previously the conclusion have been placed in the ‘Discussion’ section. The ‘Conclusion’ has been re-written.

then leave the writers opportunity to highlight what can be concluded from their overall findings. I suggest to the writers that they consider amalgamating the most important issues from the implications and current conclusions to present a sharper more concise take home message for readers.	
---	--

VERSION 2 – REVIEW

REVIEWER	Sandra Nicholson QMUL
REVIEW RETURNED	21-Jun-2020
GENERAL COMMENTS	The authors have mostly satisfactorily responded to my concerns. I would suggest that the implications section could be more concise.

VERSION 2 – AUTHOR RESPONSE

We thank Reviewer 3 for their feedback. As suggested, we have made the implications section more concise. We have proof-read the manuscript.